# Comparative Study of Heavy Metals in Selected Medicinal Plants and Extracts, Using Energy Dispersive X-ray Fluorescence

Fernando Sánchez-Lara [1,*], Eduardo Manzanares-Acuña [1], Valentín Badillo-Almaraz [1], Rosalinda Gutiérrez-Hernández [2], Karol Karla García-Aguirre [3], María Elena Vargas-Díaz [4], Álvaro Omar Hernández-Rangel [4], Karla Mariela Hernández-Sánchez [4] and Martha Celia Escobar-León [5]

1  Unidad Académica de Estudios Nucleares, Universidad Autónoma de Zacatecas, Apdo. Postal 336, Zacatecas 98000, Mexico
2  Laboratorio de Etnofarmacología Nutrición, Unidad Académica de Enfermería, Campus Siglo XXI, Universidad Autónoma de Zacatecas, Kilómetro 6, Ejido la Escondida, Zacatecas 98160, Mexico
3  Instituto Politécnico Nacional, Unidad Profesional Interdisciplinaria de Ingeniería Campus Zacatecas, Calle Circuito del Gato 202 Ciudad Administrativa, Zacatecas 99160, Mexico
4  Instituto Politécnico Nacional, Escuela Nacional de Ciencias Biológicas, Prol. Carpio y Plan de Ayala s/n, Col. Santo Tomas, Del. Miguel Hidalgo, Ciudad de México 11340, Mexico
5  Unidad Académica de Ciencias de la Tierra, Universidad Autónoma de Zacatecas, Avenida Universidad 108, Progreso, Zacatecas 98050, Mexico
*  Correspondence: fersl150789@gmail.com

**Abstract:** The use of plants has grown constantly worldwide, being a rich source of compounds that serve as established treatments for various diseases and conditions. This paper discusses the elemental composition and the level of environmental risk of heavy metals of selected medicinal plants. The extracts are investigated by energy dispersive X-ray fluorescence, a non-destructive, fast, multi-element, highly accurate and environmentally friendly analysis compared to other elementary detection methods. The studied plants *Croton dioicus* and *Phoradendron villosum* are native to Mexico. Both showed high levels of Cu and Ni, while their extracts present levels within the permissible range.

**Keywords:** heavy metals; medicinal plants; EDXRF; fluorescence

## 1. Introduction

The term heavy metal refers to any metal chemical element that has a relatively high density and is toxic or poisonous in even very low concentrations. Examples of heavy metals or some metalloids include mercury (Hg), cadmium (Cd), arsenic (As), chromium (Cr), thallium (Tl), and lead (Pb), among others [1–5]. Heavy metals are generally found as natural components of the earth's crust, in the form of minerals, salts, or other compounds. Heavy metals are dangerous because they tend to bioaccumulate in different crops. To a small degree, they can be incorporated into living organisms (plants and animals) via food and can do so through water and air as means of translocation and depending on their mobility in such media [4,6–8].

Heavy metals, so-called trace elements, can serve as micronutrients for crops, since they are required in small quantities and are necessary for organisms to complete their life cycle [9–17]. Therefore, proper establishment of health guidelines is often difficult. The content of main elements and trace elements in plants is governed both by the geochemical characteristics of the soil where it grows, as well as by the ability of plants to accumulate elements selectively.

It is important to consider that heavy metal pollution in Mexico is a problem that has been increasing mainly due to anthropic activities. Mining is known to be one of the main causes of environmental pollution by heavy metals, mainly due to the inadequate management of its waste called "mining tailings", which causes pollution problems in different states [18–20]. As a result of these increases in metal concentrations in soils due

to inappropriate practices, increased bioavailability of metals for multiple crops has been causing damage, phytotoxicity and thereby causing a latent risk to human health. The main polluting metals in Mexico, considering their toxicity and abundance are mercury (Hg), arsenic (As), lead (Pb), nickel (Ni), and copper (Cu) [6,14,21–23].

In this field, numerous chemical studies have been conducted on plants, dedicated to the determination of compounds, heavy metal content, the mechanisms of their effects, as well as the investigation of external and internal factors that influence the qualitative and quantitative content of trace elements. Most analytical methods used for the determination of metals in natural samples require the destruction of their matrix [3,24–27]. The energy dispersion X-ray fluorescence (EDXRF) method presents a technique that enables the identification of chemical elements present in multiple samples in an effective, simple and non-destructive manner for multi-elemental analysis [14,28–31].

In this sense, the objective of this study is to determine the level of environmental risk of heavy metals in the plants *Croton dioicus* and *Phoradendron villosum*—and their aqueous, ethanolic, hydroethanolic, and oily extracts—to determine the feasibility of its intake since they are used in several areas of Mexico to support the treatment of multiple diseases.

## 2. Methods and Materials

### 2.1. Study Site

The selected plants were collected in Zacatecas, México. *Croton dioicus* is a wild plant and was collected from Río Florido (23°20′39″ N, 102°59′23″ O) (Figure 1). *Phoradendron villosum* is a wild plant and was collected from Laguna del Carretero (22°16′47″ N, 102°48′18″ O) (Figure 2).

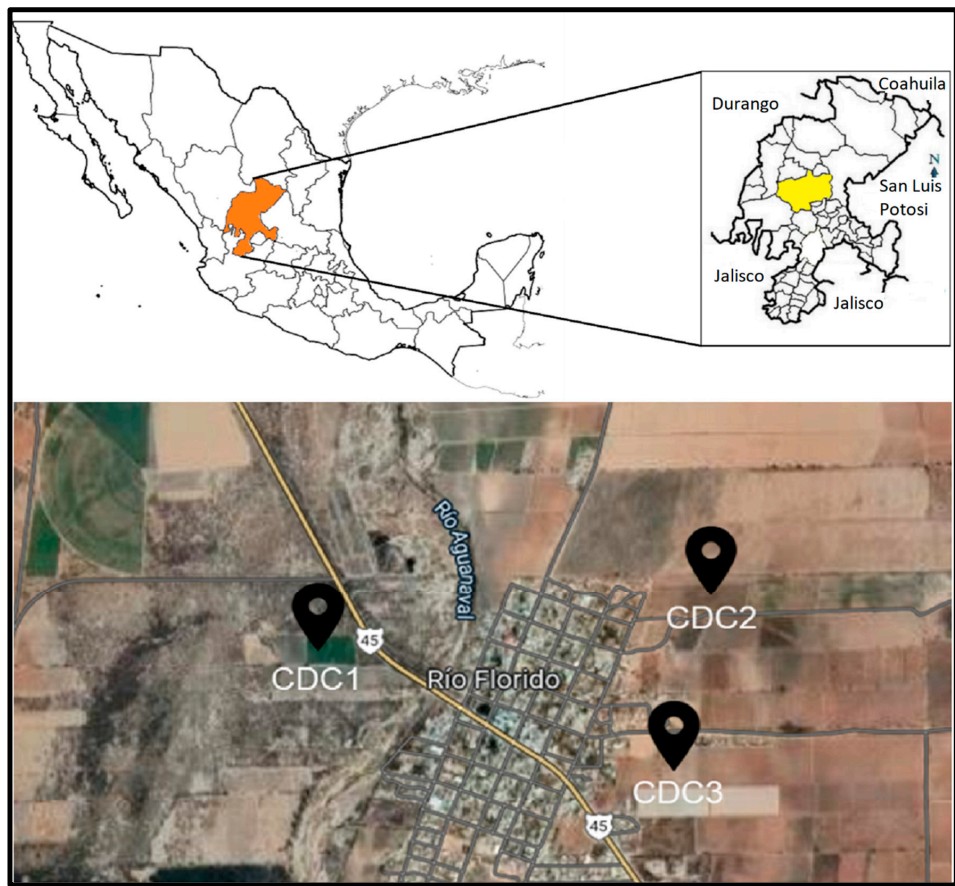

**Figure 1.** Map of the region from where the samples of *Croton dioicus* were taken: Río Florido. The samples' coordinates are $CDC_1$ (23°20′53.06″ N, 102°59′53.79″ O); $CDC_2$ (23.349884,102°59′1.87″ O); $CDC_3$ (23°20′36.60″ N, 102°59′5.89″ O).

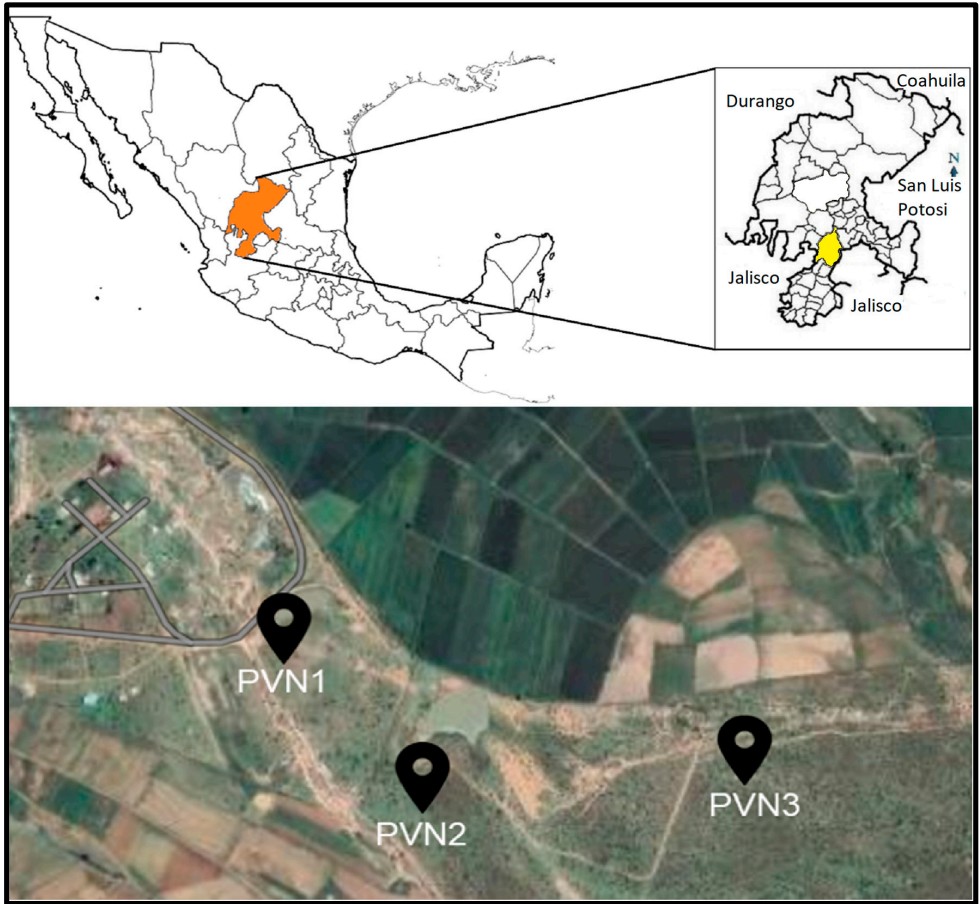

**Figure 2.** Map of the region from where the samples of *Phoradendron villosum* were taken: Laguna del Carretero. The samples' coordinates are $PVN_1$ (22°16′23.89″ N, 102°47′56.42″ O); $PVN_2$ (22°16′14.59″ N, 102°47′48.08″ O); $PVN_3$ (22°16′16.59″ N, 102°47′28.62″ O).

*2.2. Plant Identification*

The plants (Figure 3) were botanically identified and authenticated from the Herbarium of the National School of Biological Science (ENCB) of the National Polytechnic Institute by Dr. Ma. De la Luz Arreguín, head of the Laboratory of Botanical Phanerogamic. The specimens of the plants are deposited in the department for further reference. The vouchers number are CD/ZAC/2019 (*Croton dioicus*) and PV/ZAC/2019 (*Phoradendron villosum*).

*2.3. Sample Preparation*

All parts of the plants (trunk, flower, and leaves) were cleaned and washed with plenty of water to remove impurities from the environment. They were left to dry at room temperature (20 ± 5 °C) with air circulation for 3 days.

After drying, the samples were crushed in a rotary blade mill. Ground plant samples were pressed into 32 mm diameter pellets using manual press.

Some powdered samples were macerated in water, ethanol, oil (grape seed oil), and a mixture of ethanol-water in a 9:1 ratio. In the extracts of Phoradendron villosum and Croton dioicus, were macerated 520 gr and 165 gr of botanical material, respectively, in volumes of 4 L. The extracts were concentrated in a rotary evaporator. Concentrated samples were pressed into 32 mm diameter pellets using manual press.

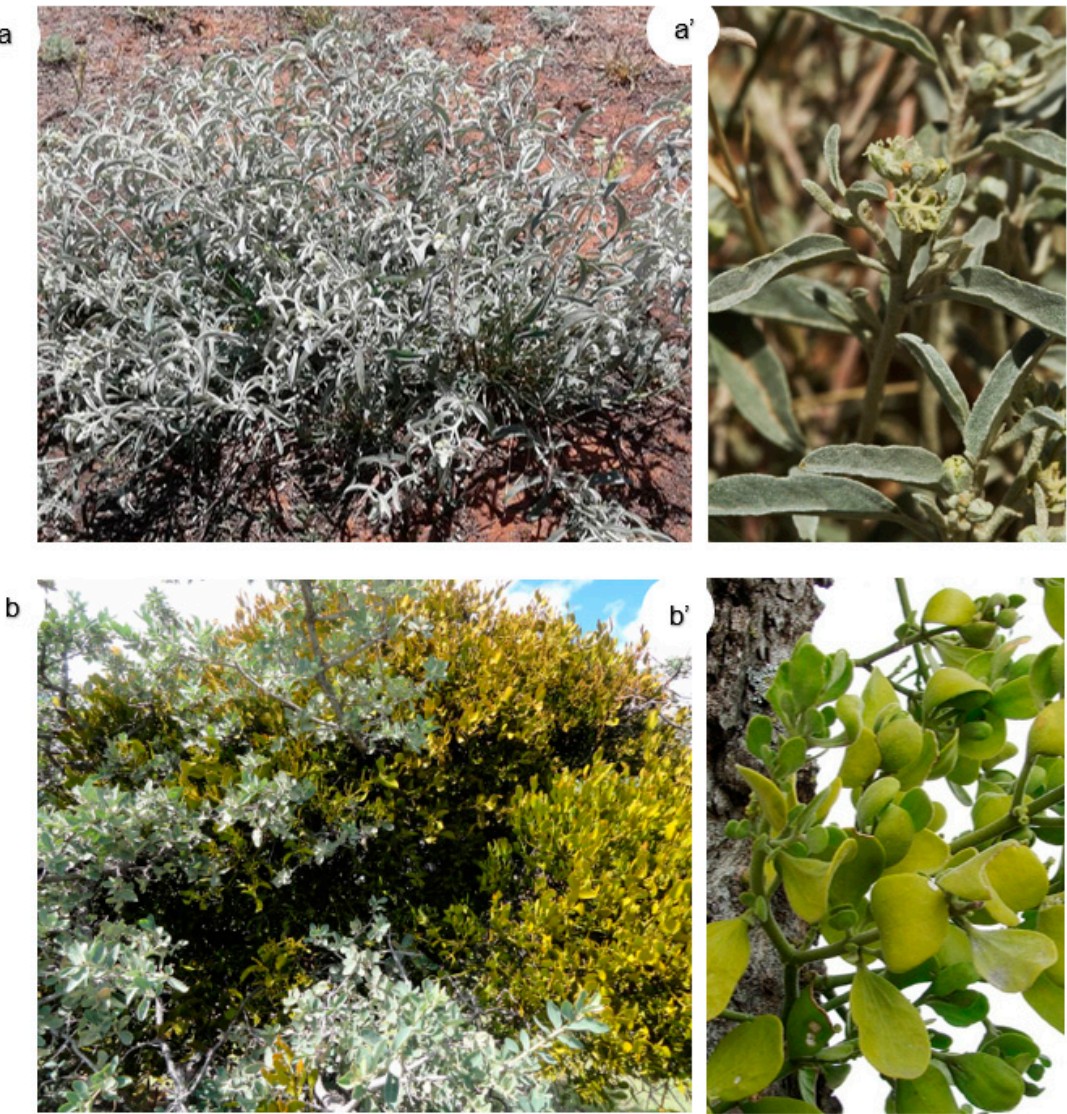

**Figure 3.** Studied plants: (**a**,**a'**) *Croton dioicus*; (**b**,**b'**) *Phoradendron villosum*.

## 2.4. Energy Dispersive X-ray Fluorescence

Pulverized plant samples and concentrated extracts were analyzed in a Mini-Pal X-ray dispersion spectrometer Philips model Pw4051 with a rhodium tube (SDD detector, detector resolution < 160 eV Mn-K$\alpha$, energy range 0–24 keV). The spectrometer was controlled by PW 4051 MiniPal/MiniMate Software V 2.0A.

The qualitative analysis of the elements is comprised between sodium and plutonium. Quantitative levels of Cu, As, Pb, and Ni were measured in three samples per plant and concentrated extracts. Each measurement was repeated five times, reporting the average. The blank and working standards were first run followed by the samples.

## 2.5. Quality Control

Prior to analysis, the calibration curves for each metal were constructed with standard samples with different concentrations. The standards used were purchased from the National Institute of Standards and Technology (NIST) and included Tomato Leaves (SRM-1573), Spinach Leaves (SRM-1570) and Apple Leaves (SRM-1515). The statistical correlation in the calibration curve for Cu, As, Ni and Pb was 0.996, 0.999, 0.999, and 0.997, respectively. The properties of calibrated graphs are shown in Table 1.

**Table 1.** Properties of calibration graphs.

| Element | EDXRFS | |
| | Equation [m] = cps/(mg/kg), [b] = cps | Regression |
|---|---|---|
| Cu | y = 0.666x + 0.011 (±0.032) | 0.996 |
| Pb | y = 0.499x + 0.064 (±0.028) | 0.997 |
| As | y = 0.633x + 0.001 (±0.001) | 0.999 |
| Ni | y = 0.700x + 0.077 (±0.059) | 0.999 |

Considering the sensitivity of assessment, the accuracy and precision of the EDXRF analysis were checked by repeatedly testing the certified reference material, Orchard leaves (SRM 1571). Five replicates were made for reference material (Cu, Pb, As, Ni), and the mean value is shown in Table 2. The results for the recoveries were between 90% and 95%, which showed a good agreement between the measured and certified values.

**Table 2.** Analytical results obtained on certified reference materials (mg/kg).

| Element (Certified SRM 1571) | Certified Value | Mean Measured Values | Recovery (%) | Accuracy (%) |
|---|---|---|---|---|
| Cu | 12 | 11.02 ± 0.70 | 91.86 | 2.14 |
| Pb | 45 | 43.82 ± 1.60 | 97.38 | 4.72 |
| As | 10 | 9.45 ± 0.48 | 94.54 | 2.23 |
| Ni | 1.3 | 1.22 ± 0.10 | 94.38 | 2.45 |

The analytical method for EDXRF quantifies the relative concentration of each element present in the sample by fluorescence [32,33].The validation studies were performed by using fundamental parameters including testing of linearity (Table 1), the limit of detection and the limit of quantification (Table 4).

The value LOD and LOQ are calculated considering the Equations (1) and (2):

$$LOD = 3.3 \frac{SD_{intercept}}{slope} \tag{1}$$

$$LOQ = 10 \frac{SD_{intercept}}{slope} \tag{2}$$

where $SD_{intercept} = S_{intercept} \sqrt{n}$ and n number of test.

### 2.6. Statistical Analysis

A one-way ANOVA was used to test for significant differences in levels of heavy metals between plants and extracts, with subsequent comparative analysis by Tukey's method. A statistically significant difference was considered to be $p < 0.05$. All data were analyzed and graphed with the software SigmaPlot 12.0.

### 2.7. Ethics Approval and Consent for Publication

All experimental protocols were approved by the Autonomous University of Zacatecas. We requested permission from the local community. The authors confirm no conflict of interest and agree with the submission of the manuscript to Applied Sciences.

### 2.8. Research and Publication Ethics

The authors confirm that the use of plants in the present study complies with international [34,35], national and institutional guidelines.

## 3. Results and Discussion

In this study, *Croton dioicus* (CDC) and *Phoradendron villosum* (PVN), as well as their aqueous, ethanolic, hydroethanolic, and oily extracts have been subjected to heavy metal analysis by EDXRF. Table 3 summarizes the uses in traditional medicine of selected plants.

**Table 3.** Uses of selected plants in traditional medicine.

| Plant Name/Family | Part Used | Common Name | Medicinal Properties |
|:---:|:---:|:---:|:---:|
| **CDC** | Whole plant | Suapatle | Hypoglycemic, purgative, anti-inflammatory |
| **PVN** | Whole plant | Oak graft | Hypoglycemic, diuretic, lung problems |

Table 4 shows the World Health Organization permissible limit (mg/kg) of Ni, Cu, As and Pb in the medicinal plant [34,35], tolerable daily intake on exposure of heavy metals for the human being based on body weight (bw) [36], LOD and LOQ of the four heavy metals.

**Table 4.** LOD, LOQ, tolerable daily intake and World Health Organization permissible limit of Ni, Cu, As, and Pb in the medicinal plant.

| Metal | Limit of Detection LOD (mg/Kg) | Limit of Quantitation LOQ (mg/Kg) | Health-Criteria Levels (mg/Kg) | Tolerable Daily Intake (mg/50–60 kg bw) |
|:---:|:---:|:---:|:---:|:---:|
| **Cu** | 0.36 | 1.09 | 10 | 3 |
| **Ni** | 0.62 | 1.90 | 1.5 | 1.4 |
| **As** | 0.01 | 0.04 | 10 | 1 |
| **Pb** | 0.41 | 1.26 | 10 | 0.25 |

It is important to determine the level of Cu, Ni, As and Pb in plants to determine the viability of their intake. This is related to the need to cope with environmental deterioration because in Mexico the issue of poisoning is important due to the operation of adjacent mines. As a result of these increases in soil concentrations of metals due to inappropriate practices, the increased bioavailability of metals to multiple crops has been causing toxicity, thereby causing a latent risk to animal and human health.

The elements detected qualitatively with the highest concentration in *Croton dioicus* and *Phoradendron villosum* are Mg, K, Ca, P, Mn, Fe, Ni, Cu and As. The elements K, Ca, and Fe [10,11] in both the plants studied, indicate a good metabolism of carbohydrates and proteins, stability in the absorption of magnesium avoiding toxicity by the same, and a balance in the development of chloroplasts and chlorine, respectively. The Mn indicates a suitable photosynthetic process. It is important to emphasize that the elements not mentioned, due to their low concentration in plants, help in their life cycle and reproduction [11].

### 3.1. Elemental Analysis

The concentrations of measured elements in the selected plants are given in Tables 5 and 6. Furthermore, the values of SD (Standard Deviations) are also in the tables. *Croton dioicus* and *Phoradendron villosum* samples are labeled CDC and PVN, respectively. In the case of extracts, they are identified with the nomenclature AQE (aqueous), AL (ethanolic), HA (hydroethanolic) and OE (oily).

**Table 5.** Analyses of *Phoradendron villosum* and extracts determined by EDXRF. BDL: below detection limit. ANOVA $p < 0.05$ * significant difference, post hoc Tukey's.

| Sample Code | Cu (mg/kg) | SD | As (mg/kg) | SD | Ni (mg/kg) | SD | Pb (mg/kg) |
|---|---|---|---|---|---|---|---|
| PVN$_1$ | 107.59 * | 0.84 * | 11.18 * | 0.34 * | 148.38 * | 1.71 * | BDL |
| PVN$_2$ | 100.48 * | 0.97 * | 14.79 * | 0.32 * | 150.43 * | 1.80 * | BDL |
| PVN$_3$ | 101.56 * | 0.72 * | 14.45 * | 0.29 * | 149.7 * | 1.76 * | BDL |
| AQE/PVN$_1$ | 3.48 | 0.29 | 0.14 | 0.04 | 3.75 | 0.13 | BDL |
| AQE/PVN$_2$ | 3.26 | 0.27 | 0.08 | 0.02 | 3.74 | 0.12 | BDL |
| AQE/PVN$_3$ | 5.1 | 0.24 | 0.1 | 0.04 | 4.14 | 0.13 | BDL |
| AL/PVN$_1$ | 4.18 | 0.32 | 0.06 | 0.03 | 3.62 | 0.25 | BDL |
| AL/PVN$_2$ | 4.25 | 0.25 | 0.07 | 0.02 | 3.14 | 0.22 | BDL |
| AL/PVN$_3$ | 4.68 | 0.33 | 0.08 | 0.02 | 4.16 | 0.26 | BDL |
| HA/PVN$_1$ | 2.9 | 0.36 | 0.16 | 0.02 | 3.2 | 0.38 | BDL |
| HA/PVN$_2$ | 3.6 | 0.22 | 0.12 | 0.02 | 2.78 | 0.23 | BDL |
| HA/PVN$_3$ | 3.57 | 0.28 | 0.2 | 0.02 | 2.96 | 0.24 | BDL |
| OE/PVN$_1$ | 3.05 | 0.22 | 0.36 | 0.03 | 2.56 | 0.22 | BDL |
| OE/PVN$_2$ | 3.18 | 0.18 | 0.24 | 0.05 | 2.77 | 0.23 | BDL |
| OE/PVN$_3$ | 3.36 | 0.15 | 0.38 | 0.05 | 2.89 | 0.17 | BDL |
| WHO's permissible limit | 10 | | 10 | | 1.5 | | 10 |

**Table 6.** Analyses of *Croton dioicus* and extracts determined by EDXRF. BDL: below detection limit. ANOVA $p < 0.05$ * significant difference, post hoc Tukey's.

| Sample Code | Cu (mg/kg) | SD | As (mg/kg) | SD | Ni (mg/kg) | SD | Pb (mg/kg) |
|---|---|---|---|---|---|---|---|
| CDC$_1$ | 76.18 * | 0.21 * | 14.79 * | 0.01 * | 102.78 * | 1.84 * | BDL |
| CDC$_2$ | 73.45 * | 0.31 * | 17.61 * | 0.01 * | 108.69 * | 1.8 * | BDL |
| CDC$_3$ | 75.19 * | 0.28 * | 14.5 * | 0.04 * | 107.24 * | 1.73 * | BDL |
| AQE/CDC$_1$ | 2.81 | 0.15 | 0.6 | 0.04 | 3.9 | 0.1 | BDL |
| AQE/CDC$_2$ | 2.41 | 0.38 | 0.47 | 0.04 | 3.39 | 0.16 | BDL |
| AQE/CDC$_3$ | 3.5 | 0.3 | 0.58 | 0.03 | 4.27 | 0.15 | BDL |
| AL/CDC$_1$ | 2.84 | 0.39 | 0.19 | 0.01 | 3.7 | 0.24 | BDL |
| AL/CDC$_2$ | 2.18 | 0.26 | 0.34 | 0.05 | 2.9 | 0.27 | BDL |
| AL/CDC$_3$ | 2.26 | 0.25 | 0.15 | 0.03 | 2.44 | 0.3 | BDL |
| HA/CDC$_1$ | 2.94 | 0.33 | 0.23 | 0.02 | 4.37 | 0.39 | BDL |
| HA/CDC$_2$ | 3.69 | 0.27 | 0.16 | 0.02 | 4.56 | 0.26 | BDL |
| HA/CDC$_3$ | 3.34 | 0.38 | 0.28 | 0.01 | 4.45 | 0.31 | BDL |
| OE/CDC$_1$ | 1.55 | 0.14 | 0.99 | 0.01 | 3.74 | 0.17 | BDL |
| OE/CDC$_2$ | 1.93 | 0.21 | 0.22 | 0.03 | 3.25 | 0.15 | BDL |
| OE/CDC$_3$ | 2.31 | 0.18 | 0.67 | 0.04 | 3.78 | 0.2 | BDL |
| WHO's permissible limit | 10 | | 10 | | 1.5 | | 10 |

### 3.1.1. Copper (Cu)

Cu is an essential trace element. It is needed for normal growth and development. A high concentration of Cu causes metal fumes fever, hair and skin discolorations, dermatitis, respiratory tract diseases and some other diseases in human beings [15].

*Croton dioicus* (CDC$_{1,2,3}$) contains a concentration of 73.45 to 75.19 mg/kg, while the *Phoradendron villosum* (PVN$_{1,2,3}$) shows a concentration of 100.48 to 107.59 mg/kg (Tables 5 and 6). WHO's permissible limit of Cu in medicinal plants is 10 mg/kg, while its intake in food is 1–10 mg/day [34]. Excess of Cu can be attributed to mismanagement of adjacent mines.

### 3.1.2. Nickel (Ni)

Ni is an essential element for plants and animals. In small quantities, Ni is necessary for the formation of red blood cells. But at a high level, it becomes toxic and causes severe diseases such as loss of body weight, loss of vision, and heart and liver failures, as well as skin irritation [28,37]. The *Phoradendron villosum* ($PVN_{1,2,3}$) plant had the highest level (148.38 to 150.43 mg/kg), while the *Croton dioicus* ($CDC_{1,2,3}$) plant had the least contents (102.78 to 198.69 mg/kg) of Ni (Tables 5 and 6). WHO's permissible limit of Ni in medicinal plant is 1.5 mg/kg, while its routine requirement for mankind is 1 mg/day [34]. The high level of Ni can be attributed to waste misuse in mines adjacent to the region.

### 3.1.3. Arsenic (As)

Arsenic at very low concentrations stimulates the growth of the plant. This is not essential for the growth of the plant, and crop yields decrease at high concentrations. The main effect of arsenic on plants appears in the destruction of chlorophyll. In humans, immediate symptoms of acute arsenic poisoning include vomiting, abdominal pain and diarrhea.

The $CDC_{1,2,3}$ samples had 14.50 to 17.61 mg/kg of As, while the $PVN_{1,2,3}$ content is 11.18 to 14.79 mg/kg (Tables 5 and 6). WHO's permissible limit of As in medicinal plants is 10 mg/kg, while its intake in food is generally well below 1 mg/kg [34].

### 3.1.4. Lead (Pb)

Lead affects multiple body systems, including the cardiovascular and neurological, hematological, digestive and renal systems. Even relatively low levels of exposure can cause severe and irreversible neurological damage in some cases. WHO's permissible limit of Pb in medicinal plants is 10 mg/kg [34]. For all analyzed samples, the Pb content value is below the detection limit (Tables 5 and 6).

### 3.2. Extract Analyses

The metal study was conducted to see what percentage passes to each extract, analyzing the viability of ingestion avoiding toxicity and its possible damage to health, owing the plant is not consumed directly.

The samples analyzed contained a significantly lower amount of Cu compared to the standard reference material, NIST 1571 Orchard leaves (Figure 4). The Cu content in *Phoradendron villosum* passes to the extracts between of 2.5 to 4.8% and the extracts of *Croton dioicus* in a range of 2 to 5% (Tables 5 and 6). In the case of Cu, it can be consumed up to 3 mg/day (Table 4). Considering the tolerable daily intake, the extracts $AL/PVN_{1,2,3}$, $AQE/PVN_{1,2,3}$, $HA/PVN_{1,2}$, and $OE/PVN_{1,2,3}$ exceed the limit.

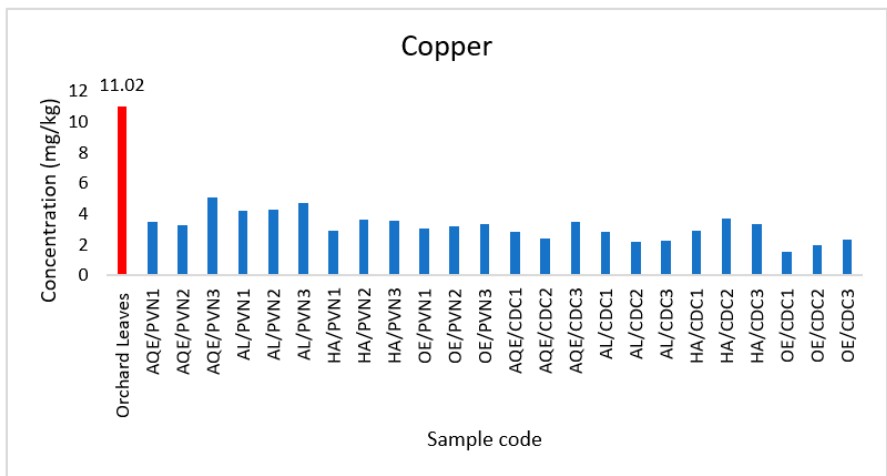

**Figure 4.** The graph shows Cu concentrations (mg/kg) in extract samples with standard reference material NIST 1571 Orchard leaves.

Levels of Ni in the extracts of *Phoradendron villosum* passed a percentage between 1.6% to 3% and Ni at *Croton dioicus* were between 2.2% to 4.2% (Tables 5 and 6). The content of Ni in the extracts for both plants is higher than the standard reference material NIST 1571 (Figure 5). Daily consumption is 1.4 mg/day, exceeding by approximately 1 to 3 mg/kg depending on the extract. For the As, it is recommended to consume less than 1 mg/day (Table 4), existing in the extracts of both plants in levels less than 0.6 mg/kg, except the samples OE/CD$_{1,3}$. The samples analyzed contained a significantly lower amount of As compared to the standard reference material NIST 1571 (Figure 6). For extracts analyzed, the Pb content value is below the detection limit (Tables 5 and 6). The concentration of Pb is higher in the reference material NIST 1571 than in the samples from both plants (Figure 7).

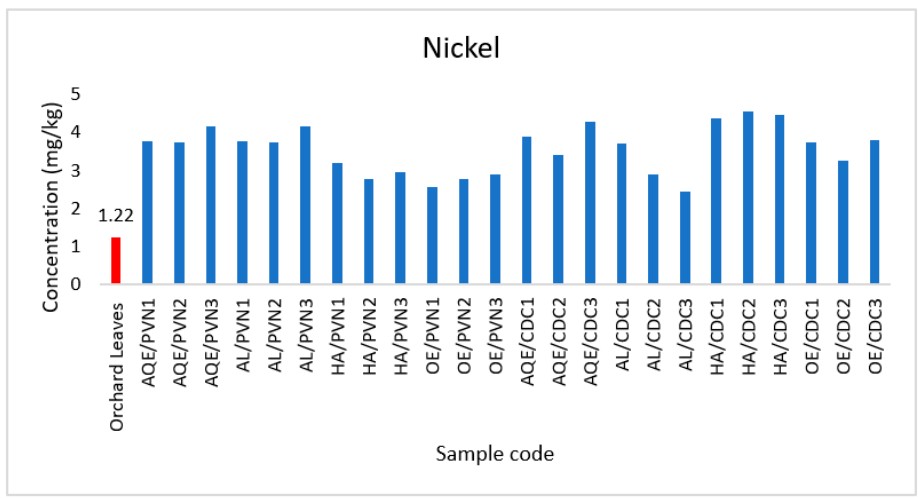

**Figure 5.** The graph shows Ni concentrations (mg/kg) in extract samples with standard reference material NIST 1571 Orchard leaves.

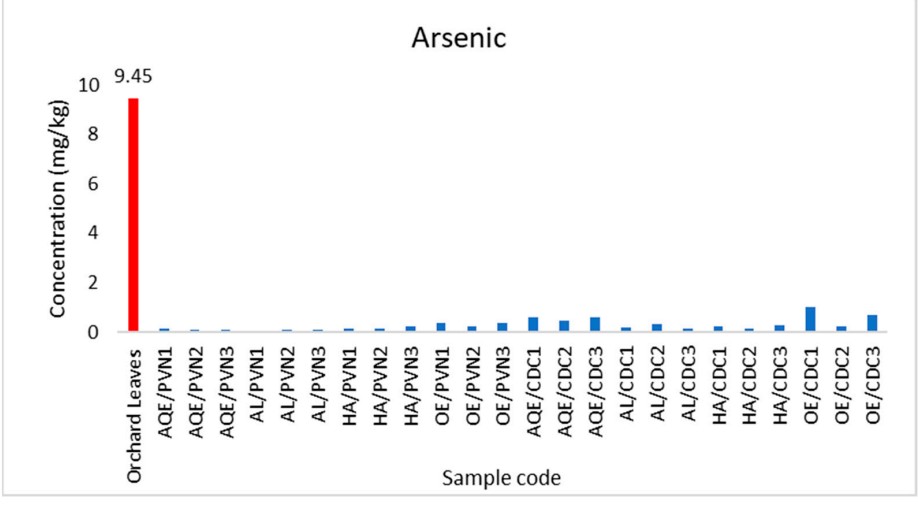

**Figure 6.** The graph shows As concentrations (mg/kg) in extract samples with standard reference material NIST 1571 Orchard leaves.

The extracts of the investigated plants are consumed in doses of 0.5 to 2 mL of extract dissolved in 200 mL of water, having even less risk of consuming significant amounts of heavy metals. Considering the form of consumption is avoided to a great extent, the intoxication by Ni, being even the levels that pass to the extracts, is considerably low. It is observed that the values of heavy metals in the extracts in *Phoradendron villosum* and *Croton dioicus* pass in different percentages depending on the solvent used, lowering the levels,

and considerably avoiding possible toxicity when consumed with supportive treatment in some conditions.

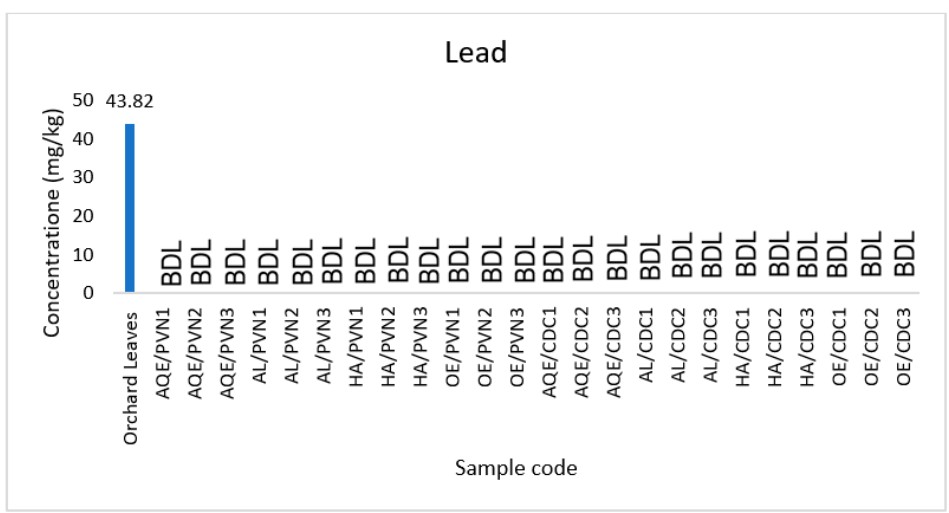

**Figure 7.** The graph shows Pb concentrations (mg/kg) in extract samples with standard reference material NIST 1571 Orchard leaves.

In addition, no significant difference in the statistical study by ANOVA and Tukey's method was found in the levels of heavy metals between the extracts (Tables 5 and 6). However, there is a marked statistical difference between the content of the heavy metals studied in the plants and what remains in each extract after maceration.

## 4. Conclusions

The EDXRF method was efficiently employed for measuring Cu, As, Ni, and Pb elements in the medicinal plants and their aqueous, ethanolic, hydroethanolic, and oily extracts.

It is also of note to indicate that the uptake of any or all of elements measured in this study will depend on the method of administration of these medicinal plants. It is important to mention that the extracts should be diluted before consumption, as otherwise one does risk heavy metal intake of certain elements above WHO limits. The present study concludes that the number of toxic elements that are investigated in the medicinal plants studied are not harmful to human health. Finally, this research will be very useful for pharmacologists who would like to pursue further study in the alternative medicines.

**Funding:** This research received no external funding.

**Institutional Review Board Statement:** Not applicable.

**Informed Consent Statement:** Not applicable.

**Data Availability Statement:** All data generated and analyzed during this study are included in this published article.

**Conflicts of Interest:** The authors declare no conflict of interest.

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
