# Peer review of "Comparative Study of Heavy Metals in Selected Medicinal Plants and Extracts, Using Energy Dispersive X-ray Fluorescence"

_applsci, doi:10.3390/app122211772_

Round 1

Reviewer 1 Report

The review has been included as pdf file.

Author Response

First of all, I thank you for each of your comments as they enrich our research.

We have considered each of your recommendations and added some others.

1. Ordered references using Mendeley.  

2. The introductory section was changed in its entirety, omitting the information about the generalities of EDXRF.

4. The objective was refocused.

5. The figure with the calibration curves was omitted, leaving the information of the calibration equations with their respective errors.

6. The supplementary files were deleted as they are not important for the manuscript and do not fit to the text.

7. Regarding calibration curves we made a mistake by attaching a wrong one in the first shipment.

8. The procedure for calculating LOD and LOQ is included.

9. Daily tolerable intake values are added to Table 3.

10. New references were included.

11. The units of the calibration curve variables in table 1 are added.

I send greetings.

Reviewer 2 Report

Sánchez-Lara et al. report the use of Energy-dispersive X-ray fluorescence (EDXRF) analytical technique for checking the elemental composition and heavy metal contents in the medicinal plants Croton dioicus Cav., Phoradendron villosum Nutt., and its extracts. Overall, the work is interesting and has some important elements, especially in terms of the measuring the concentration of Cu, As, Ni, and Pb elements in the medical plants, which could generate several disorders even in low concentrations.  Once the following specific points listed below are addressed, I believe that the manuscript is potentially publishable in Applied Sciences and should be further considered:

11.  The order of the references must be arranged. You can use Mendeley to help you with this.

22.  The introductory section must be reorganized. It should look like one unit, one text that corresponds to the topic. In your work, it looks like a set of separate sentences where each sentence is a new line. Please correct it.

33.   General information about EDXRF spectrometric analysis in the introduction part should be omitted because it is common knowledge.

  4. The Energy-dispersive X-ray fluorescence (EDXRF) analytical technique is not novel from a materials characterization perspective. Therefore, the novelty of the manuscript should be further elaborated.

55.  The caption of Figure 3 must be changed as it is illegible.

   6.  Why did you measure only 4 points for Pb when you have formed the calibration curve (Figure 4)?

  7. Please remove supplementary files, as they are not important for the manuscript and do not fit to the text.

Author Response

First of all, I thank you for each of your comments as they enrich our research. 

We have considered each of your recommendations.

1.  Ordered references using Mendeley.

2. The introductory section was changed in its entirety, omitting the information about the generalities of EDXRF.

4. The objective was refocused.

5. The figure with the calibration curves was omitted, leaving the information of the calibration equations with their respective errors.

6. The supplementary files were deleted.

7. Regarding calibration curves we made a mistake by attaching a wrong one.

I send greetings.

Reviewer 3 Report

1/ The authors of plant names (abbreviated author names) should only be provided in the Methods. Just Latin binomilas should be used in other parts of the text.

2/ Methods: geological bedrock at sampling sites must be indicated.

3/ Methods: are herbarium IDs available for the deposited plant specimens?

4/ Table 2. Certified values must be presented with reported uncertainty.

4/ Table 2. The mean measured values and accurracy do not correspond with reality. With highly precise ICPQQQMS, the acurracy can be expressed with 3 significanbt digits, e.g. 13.8 mg/kg, not better. By using XRF, the acurracy is surely much worse. The presented results like 43.822 ppm for Pb are nonsense. 44 +/- mg/kg ppm would be more realistic. Similar comment is applied to all parts of the manuscript. 148.386 ppm Ni determined by XRF is nonsense, such acurracy would hardly to be achived even by using isotopic dilution method.

5/ "The qualitative elements detected in Croton dioicus Cav. are Mg, K, Ca, Mn, Fe, Ni, Cu, and As. The elements in Phoradendron villosum Nutt. are Mg, P, Cu, Ni, K, Fe, Ca, As, and Mn." I do not really understand meaning of this part, needs editing.

6/ Cu and Ni concentrations in the analyzed plants are surprisingly high, do not correspond to concentrations commonly reported for vascular plants. Is the data reliable? How do the authors explain the high levels? They do not even comment on them. For example, the Ni concentration in plant leaves ranges from 0.05 to 5 ppm. But the authors report values exceeding 100 ppm! This must be discussed in the text.

7/ Conclusion. The first paragraph should be deleted. It is not a conclusion.

8/ Conclusion. Diabetes mellitus is mentioned but no research has been done on this disease, which was also mentioned in the Abstract. Why? Delete this from the whole manuscript.

9/ I have serious doubts about the analytical data. The authors used SRM 1571 for quality control but also as standard. I would like to see better quality control in this study.

10/ "The elements present in medicinal plants are Cu, Ni, K, Fe, Ca, As, Ni, Mn, and Mg." This sentence in the Abstract makes no sense. The whole periodic chart is present in plant tissues. It is just a matter of concentration.

Author Response

I thank you for your comments, as they enrich our research.
We have considered the changes you have highlighted. 

I attach the article with the relevant corrections and at the end I add the identification vouchers.

Round 2

Reviewer 1 Report

Comments have included as word file.

Author Response

Good afternoon, 

I greet you again, we have considered the points of your review, correcting some aspects and adding others, according to the recommendations. I appreciate your comments. 

1.- We perform calibration of the equipment to correct the calibration curves and consequently some statistical parameters.
2.- We consider the measurement of three samples for each extract and plant. 
3.- We add precision calculation.
4.- We add more references according to your recommendation.

Regards.
